

# Downstream development associated with two types of ridging South Atlantic Ocean anticyclones over South Africa

Thando Ndarana[1], Lesetja E. Lekoloane[2], Tsholanang S. Rammopo[1], Chris J. C. Reason[3], Mary-Jane M. Bopape[2], Hector Chikoore[4], Francois A. Engelbrecht[5]

[1]Department Geography, Geoinformatics and Meteorology, University of Pretoria, Hatfield, South Africa
[2]Department of Research and Innovation, South African Weather Service, Centurion, South Africa
[3]Department of Oceanography, University of Cape Town, Cape Town, South Africa
[4]Unit for Environmental Sciences and Management, North-West University, Vanderbijlpark, South Africa
[5]Global Change Institute, University of the Witwatersrand, Johannesburg, South Africa

*Correspondence to*: Thando Ndarana (thando.ndarana@up.ac.za)

**Abstract.** There are at least two types of ridging South Atlantic Ocean high pressure systems in the South African domain. Type-N events occur north of 40°S and Type-S occur south of this latitude line. This study shows that there is no evidence of surface downstream development in terms of the evolution of eddy kinetic energy and associated ageostrophic geopotential fluxes for both types of ridging high events. Rather, for these systems downstream development is an upper level process. The

baroclinic waves associated with the ridging develop from baroclinic instability, by converting eddy available potential energy to eddy kinetic energy. The bulk of the conversion is located at the upstream end of the waves. The downstream trough, which is the part of the wave that influences upward motion over South Africa, develops from the transport of eddy kinetic energy across the trough axis by means of ageostrophic geopotential fluxes. These fluxes are stronger for Type-S events. The absence of downstream development at the surface and the presence of it aloft demonstrates that there are differences in the underlying

dynamics in the evolutions of these systems in the vertical. The evolution of eddy kinetic energy associated with baroclinic waves that occur during the ridging events is different from what has been observed for cut-off low pressure systems in the South African domain.

## 1. Introduction

The subtropical South Atlantic Ocean is characterised by a quasi-stationary anticyclonic circulation that exhibits pronounced

variability in intensity and position at intra-annual and inter-annual (Sun et al., 2017) and multi-decadal (Reason, 2000) time scales. It has also been shown that it might be impacted by future climate change (Engelbrecht et al., 2009; Reboita et al., 2019) and therefore expected to change at decadal and centennial time scales. Under certain stratospheric conditions (Ndarana et al., 2018), this anticyclone extends east and ridges around the South African landmass to induce an influx of moisture into the country (Cook et al., 2004; Dyson, 2015; Engelbrecht et al., 2015; Ndarana et al., 2021a). This eastward extension process

is referred to as ridging. Ridging highs contribute to the moisture budget of southern Africa, with the South Atlantic and Indian Oceans being important moisture source regions (Rapolaki et al., 2020). These oceans are not the only sources of moisture





for South Africa. Water vapour originates from the Equatorial South Atlantic Ocean (Vigaud et al. 2007) and Equatorial Indian Ocean (D'Abreton and Tyson 1995) and transported into tropical Africa. The Angola low (Rouault et al. 2003; Reason and Jagadheesha 2005; Crétat et al. 2018; Howard and Washington 2018) then organises it in a south easterly direction (Hart et al.

2010) and tropical/extra-tropical interaction events then transport the moisture further south (Ratna et al. 2013; Macron et al. 2014).

Ridging highs are consequently important in terms of the dynamics of rain-producing weather systems of South Africa because they are low level processes that co-occur with moving troughs (Piva et al. 2008; Ndarana et al. 2022) that may develop into cut-off low (COL) pressure systems (Taljaard 1985; Singleton and Reason 2007) when the waves break (Ndarana and

Waugh 2010; Reyers and Shao 2019; Barnes et al. 2021). Heavy rainfall that sometimes leads to floods which cause extensive damage to property and loss of life often results from ridging highs that occur in tandem with COLs (Trigaardt et al., 1988; Singleton and Reason, 2006). Moreover, the ridging highs may also be associated with tropical-temperate troughs (Hart et al. 2010) whose characteristic outgoing longwave radiation anomaly pattern (Fauchereau et al. 2009) intensifies and reaches maximum values as the ridging process matures (Ndarana et al. 2021a). It follows then that understanding the dynamics

associated with this process at both the lower and upper levels may play an important role in informing predictability studies of extreme rainfall events in the South Atlantic Ocean/South African sector. One approach that previous studies have employed in understanding upper level dynamical processes is the local energetics framework to diagnose the evolution of baroclinic waves (Orlanski and Sheldon, 1995), which gives rise to the notion of downstream development (Orlanski and Sheldon, 1993, 1995). The local energetics framework has since been applied to a broad range of atmospheric dynamical settings, beyond the

evolution of baroclinic waves, including intensifying jet streaks (Lackmann et al., 1999), surface cyclolysis (McLay and Martin, 2002), life cycles of conservative extratropical cyclones that exhibit explosive deepening behaviour (Decker and Martin, 2005), extratropical transitions of tropical cyclones (Harr and Dea, 2009), travelling mid-tropospheric troughs (Piva et al., 2010), the evolution of cut-off low (COL) pressure systems (Gan and Piva, 2013; 2016; Ndarana et al., 2021b; Pinheiro et al., 2021), dynamical processes that lead to the formation of frost events (Müller et al., 2017) and the processes that maintain

jet wave trains (Li, 2021). It might therefore find some applications to ridging pressure systems as well.

Ndarana et al. (2022) showed that there are at least two types of ridging high evolution that occur in the South African domain. These were categorised according to which latitude they occur, relative to the 40°S latitude line. Type-S (Type-N) ridging events occur south (north) of this latitude line. They also showed that the different types of ridging highs are associated with baroclinic waves aloft, whose troughs pass over South Africa as the ridging process is initiated. Whilst other studies (e.g.

O'Brien and Reeder, 2017) have analysed the evolution of these waves in the South Atlantic Ocean/South African sector using their interaction with the jet as a basis for their analysis; little is known about the dynamical characteristics of these waves from a downstream development perspective. Recent studies (e.g. Ndarana et al. 2021b; Pinheiro et al. 2021) showed that COLs are characterised by such downstream development, which is facilitated by Rossby waves that break east of the Greenwich Meridian in an anticyclonic fashion (Thorncroft et al., 1993). The breaking waves facilitate ageostrophic

geopotential fluxes that draw eddy kinetic energy from the mid-latitudes into the COL's formation region, which in turn





induces the closed cyclonic circulation of the COL, together with the effect of the high potential vorticity anomaly that does the same (Hoskins et al., 1985). It is well known from forecasting experience in southern Africa that COLs are often accompanied by ridging highs at the surface, and that not all ridging highs have a COL aloft associated with them (Engelbrecht et al., 2015). There are many instances when a ridging high occurs with an ordinary trough aloft. Whilst these complex relations

still need to be quantified; the fact that COLs exhibit particular downstream development characteristics (Ndarana et al. 2021b; Pinheiro et al. 2021) means that the ridging highs associated with them may be characterised by the same type or structure of the evolution of the eddy kinetic energy. However, it is still an open question whether for the general case, that is, the case when a general trough is associated with the ridging high would behave as the COL or not. Moreover, the differences in the behaviour of the baroclinic waves that were identified in Ndarana et al. (2022) have not been extended to the associated

dynamical processes as yet. Using the local energetics framework (Orlanski and Katzfey 1991), this study aims to

- further show that the baroclinic waves that are associated with Type-N and Type-S ridging events are dynamically different, and
- show that downstream development associated with COLs is different from that observed for baroclinic waves that pass over South Africa during the evolution of ridging highs.


The rest of the paper is structured as follows: in the next section a brief review of the energetics diagnostic framework is presented, followed by Data and Methods in Section 3. The results are presented in Section 4 and the Concluding remarks are given in Section 5.

## 2. Review of flow processes and downstream development in baroclinic waves

To characterise downstream development, the variables of interest are first divided into their basic state and perturbation parts, where the former are represented by capital letter symbols with a subscript $m$ and the perturbation fields are the small letters so that the instantaneous horizontal velocity and geopotential fields are separated into their basic state and perturbation portions as $\mathbf{V} = \mathbf{V}_m + \mathbf{v}$ and $\Phi = \Phi_m + \phi$, respectively. Note that the horizontal flow may be decomposed into its zonal and meridional components as $\mathbf{V} = U\mathbf{i} + V\mathbf{j}$ and that the three dimensional flow is represented by $\mathbf{U} = \mathbf{V} + \omega\mathbf{k}$, where $\omega$ is the

vertical velocity in pressure coordinates, so that $\mathbf{u}$ becomes its perturbation. Note that $\mathbf{i}$, $\mathbf{j}$ and $\mathbf{k}$ are unit vectors in the zonal, meridional and vertical directions, respectively. On this basis, the eddy kinetic energy is defined as $K_e = \frac{1}{2}\mathbf{v} \cdot \mathbf{v}$ and the flux form (Orlanski and Sheldon 1993; McLay and Martin 2002) of the equation that describes its evolution is then given by

$$\partial_t K_e = -\nabla_p \cdot (\mathbf{V}K_e) - \partial_p(\omega K_e) - \mathbf{v} \cdot \nabla_p \phi + \left[\mathbf{v} \cdot (\mathbf{u} \cdot \nabla_p)\mathbf{V}_m\right] + \text{Residual} \qquad (1)$$


The first and second terms on the right hand side of Eq. (1) are the horizontal and vertical eddy kinetic energy flux convergence, respectively. The third term can be interpreted as the advection of the perturbation geopotential field by the horizontal



perturbation velocity field. The fourth term is the Reynold's stress term, which is the conversion of $K_e$ to mean kinetic energy and the "Residual" is obtained by subtracting the terms just described from $\partial_t K_e$. The advection term, $\mathbf{v} \cdot \nabla_p \phi$, plays a

dominant role in this eddy kinetic energy budget equation and various processes are embedded in it (Orlanski and Katzfey 1991). These processes are most clearly seen when it is decomposed in the following manner,

$$- \mathbf{v} \cdot \nabla_p \phi = -\omega \alpha - \nabla_p \cdot (\mathbf{v}\phi)_a - \partial_p(\omega\phi) \tag{2}$$

Here $\alpha = -\partial_p \phi$ and the rest of the symbols have already been defined. Once decomposed, it becomes clear that $-\mathbf{v} \cdot \nabla_p \phi$ is a dominant term in Eq. (1) because it embodies all the $K_e$ generation processes. The first term on the right-hand side $-\omega\alpha$ is the baroclinic conversion from eddy available potential to eddy kinetic energy. There clearly needs to be vertical motion and it is essentially a vertical heat flux (Li., 2021). The second term is the convergence of ageostrophic geopotential flux of $K_{e,}$ which is a horizontal process and the fluxes themselves, $(\mathbf{v}\phi)_a$, as represented by the red curved arrows in Fig. 1, serve to

radiate energy in the direction of the ageostrophic flow, across a ridge axis, so that a downstream $K_e$ may then develop at the inflection point immediately east of the ridge axis. Across a trough axis, the downstream directed ageostrophic flux is caused by the fact that the flow is subgeostrophic (with a ageostrophic flow directed upstream) and $\phi < 0$. The term $\partial_p(\omega\phi)$ is the vertical geopotential flux convergence.

Based on Orlanski and Sheldon (1995) for baroclinic waves in general and for the COL case in the South Atlantic

Ocean/southern African sector (Ndarana et al., 2021b) and Southern Hemisphere in general (Pinheiro et al., 2021), downstream development is a three stage process. During the first stage a $K_e$ centre I is generated by means of baroclinic conversion ($-\omega\alpha > 0$), as represented by the red hatched oval shape. At the bottom of the wave, across the ridge axis, the supergeostrophic flow, as represented by the downstream directed ageostrophic flow, combined with $\phi > 0$, combine to produce the ageostrophic geopotential fluxes that begin to transport $K_e$ downstream. As $K_e$ I decreases in strength from state 1 to stage 2,

$K_e$ II increases in strength to reach a maximum during the latter. This is largely informed by the increase in positive perturbation geopotential in the ridge and the ageostrophic flow; thus increasing transport of energy downstream via the ageostrophic geopotential fluxes, which are particularly strong in COLs. This strength is caused by anticyclonically breaking waves which induce deep geopotential anomalies and a strong supergeostrophic flow ahead of the closed circulation (Ndarana and Waugh, 2021; Reyers and Shao, 2019; Ndarana et al. 2021b). The negative perturbation field across the trough axis (downstream of

energy centre II) begins to appear and the upstream ageostrophic flow lead to energy transfer toward the inflection point beyond the trough axis to cause $K_e$ III to develop. As was the case with the strengthening ridge, as the trough deepens, the flux increases; thus increasing $K_e$ centre III.



## 3. Data and Methods

### 3.2 Data

We use the ridging high cases that were identified objectively in Ndarana et al. (2022). As in that study, to identify ridging South Atlantic Ocean high pressure systems, we use the Fifth Generation European Centre for Medium-Range Weather Forecasts Reanalysis (ERA5, Hersbach et al., 2020) products from 1979 to 2020. The horizontal grid spacing of each dataset is 2.5° at 6-hourly time intervals. Even though these products are available at finer mesh grids than this, we deem the chosen grid spacing sufficient because ridging high pressure systems are synoptic scale processes, the horizontal scale of which is ~$10^6$ m (Holton and Hakim, 2014). The variables used are mean sea level pressure (MSLP), the zonal and meridional wind components and geopotential heights to calculate the geostrophic $(v_g \mathbf{i} + v_g \mathbf{j})$ and ageostrophic $(v_a \mathbf{i} + v_a \mathbf{j})$ wind fields. In this study we assumed a variable $f$ geostrophic flow (Blackburn 1985) so that it is not non-divergent (Cook 1999).

### 3.2 Methods

Ridging highs are identified using a simple algorithm consisting of three steps. Its details are provided in Ndarana et al. (2018) and only a brief description is provided here. In the first step, closed contours in the 6-hourly MSLP fields are identified within a domain bounded by 40°W and 60°E. We then group these closed contours so that concentric contours in the South Atlantic Ocean represent the South Atlantic Ocean High ( SAOH). This is the second step. In the third and final step, if the outermost contour extends east of the 25°E longitude line, we record such instances as the ridging process having occurred. If this condition is met on consecutive time steps, without any breaks in between, then this constitutes a single ridging event and is the basis on which the duration of the events is determined.

To establish the climatological behaviour of the ridging events and associated fields, composite calculations are used. Only the duration and location of events, relative to the 40°S latitude line are taken into consideration and assumed. That is, we composite together events of durations 24 hours or longer, that occur north or south of this latitude line. No other assumption is made about the fields. This means that their average structures appear in a natural fashion, signalling their climatological presence during the evolution of ridging events. The basic state (all the variables with subscript $m$) is calculated by the taking a 31-day mean centred on time step at which the South Atlantic Ocean high pressure system begins to extend east, to ridge across South Africa. What this implies is that the basic state variables are constant right through evolution of the ridging high. The perturbation variables are then obtained by subtracting the basic state variables from the total variables.

We also use composite means to analyse the synoptic climatological behaviour of these systems and the associated variables. Unlike in the case of COLs where the systems and associated fields are brought together so that the centres of the closed circulation coincide, we composite the fields on the basis of duration only and do not make any attempt to shifte the events so that they coincide in space. This, of course, is influenced by the fact that the ridging of the SAOH is a quasi-stationary process that varies in position, mostly latitudinally (Ndarana et al. 2022). In this case the ridging highs that last longer than 24 hours are brought together such that the time of their inception coincide and then averaged. Details of the compositing





procedure are provided in Ndarana et al. (2022). All the variables that have been calculated according to the diagnostics reviewed in Section 2 are then averaged

## 4. Results

### 4.1 Vertical profiles of the energetics diagnostics

Unlike previous studies on energetics of baroclinic waves (e.g. Orlanski and Katzfey, 1991; Orlanski and Sheldon, 1995; Lackmann et al., 1999; McLay and Martin, 2002; Decker and Martin, 2005; Danielson et al., 2006; Harr and Dea, 2009; Piva et al., 2010; Gan and Piva, 2013; 2016; Ndarana et al., 2021b; Pinheiro et al., 2021) who integrated the diagnostics vertically, we consider vertical variations of the processes associated with ridging high eddy energy exchange. The reason for this is that not only does the flow at the surface differ during the evolution of a ridge, there are also variations between types of ridges as shown in Ndarana et al. (2022). Moreover, there are substantial differences in flow between the surface and the middle to the upper troposphere. To address the differences between the characteristics of downstream development that is associated with the two types of ridging events, we make use of the eddy kinetic energy framework (see Eqs. (1) and (2)) and the perturbation fields as described in Section 2.

Fig. 2 shows all the diagnostics under consideration averaged for the composite mean fields from t = 0 to +48 hours in the domain bounded by 30°S, 60°S, 30°W and 60°E to the north, south, west and east, respectively. The red and black (green and blue) curves represent positive (negative) values for Type-N and Type-S ridging events, respectively. The vertical profiles of $K_e$ (Fig. 2a) show that it increases as a function of pressure level until it attains a maximum at about 250 hPa. This vertical increase in $K_e$ is caused by the fact that $|\mathbf{v}|$; the magnitude of the perturbation flow, also increases in the vertical to become strongest at that level as well (Figs 2b and c). The latter, of course, is caused by the thermal wind relation, which is valid for the perturbations fields. Furthermore, the $K_e$ associated with Type-S ridging events is stronger than that associated with Type-N events. Figs 2b and c show that the source of this difference is the zonal component of the perturbation flow being stronger for the former.

The tendency of $K_e$ is also a maximum aloft at levels slightly lower than 250 hPa (Fig. 2d). This may be influenced by many factors, two of which are baroclinic conversion from eddy available potential energy and the convergence/divergence of the ageostrophic geopotential fluxes. The former is informed by the vertical circulation, which attains maximum values at about 600 hPa, as shown in Fig. 2e. Note that the upward vertical motion (i.e. $\omega < 0$) that is associated with Type-S events is stronger. Fig. 2f shows that the vertical circulation maximises where $\nabla \cdot \mathbf{v}_a$ is a minimum. This is a well-known phenomenon. Combined with specific volume profiles shown in Fig. 2g, this leads to baroclinic conversion from eddy available potential energy to eddy kinetic energy that is a maximum at about 400 hPa (Fig. 2 (h)). The ageostrophic geopotential flux has been shown in previous studies to transport energy downstream in a baroclinic wave so that downstream development occurs (Orlanski and Katzfey, 1991; Orlanski and Sheldon, 1995; Lackmann et al., 1999; McLay and Martin, 2002; Decker and Martin, 2005; Danielson et al., 2006; Harr and Dea, 2009; Piva et al., 2010; Gan and Piva, 2013; 2016; Ndarana et al., 2021b;





Pinheiro et al. 2021). Fig. 2i clearly shows that downstream development that is associated with ridging highs in the South Atlantic Ocean/South African sector is largely an upper level process, as the field $|\nabla \cdot (\mathbf{v}_a \phi)|$ is a maximum at 250 hPa, where flow divergence is a maximum. In general, the vertical profiles of $\nabla \cdot (\mathbf{v}_a \phi)$ are informed by those of flow convergence in Fig. 2f and the ageostrophic fluxes by the flow components in Figs 2b and c. Combined with the fact that there are significant differences between the dynamical processes aloft and lower levels during ridging (as shown in Ndarana et al., 2022), Fig. 2

suggests that there might be need to consider the energetics for the upper and lower levels, separately, as will be done in the following sections.

### 4.2 The characteristics of lower level energy during ridging highs

We now consider the horizontal evolution of $K_e$, first at the surface in this section and subsequently for the upper troposphere, in order to characterise the downstream development associated with Type-N and Type-S events at these levels. To link the structure of the MSLP to the behaviour of $K_e$ and the differences in that behaviour between the two types of ridging events at the surface, we use the connection between the perturbation geopotential field, which is clearly induced by the ridging process (Ndarana et al., 2022), and the associated horizontal perturbation flow, in terms of which the $K_e$ is defined. Scaling arguments

from standard texts on dynamic meteorology (e.g. Holton and Hakim 2014) have shown that in the mid-latitudes, the synoptic scale flow is in approximate geostrophic balance (i.e. $\mathbf{V} \approx f^{-1} k \times \nabla_p \Phi$). The mean flow may also be characterised in the same way, so that $\mathbf{v} \approx f^{-1} k \times \nabla_p \phi$ is , thus suggesting a direct link between the horizontal perturbation flow and the horizontal gradient of the perturbation geopotential field. Fig. 3 shows the 1000 hPa composite mean fields of $K_e$ (shaded), superimposed with a few selected MSLP (thick dashed black) contours, the perturbation geopotential field $\phi$ and associated

horizontal perturbation geostrophic flow $\mathbf{v}_g$, represented by the grey arrows for DJF. It is clear from this figure that the surface horizontal flow that is important during the evolution of ridging high pressure systems occur in the southern edge of the subtropical belt and the mid-latitudes, where the geostrophic approximation is valid because the Rossby number $R_o = U/fL$ is small enough there. It follows then that $\mathbf{v} \approx \mathbf{v}_g$, so that the differences in the strength of the flow that is associated with the two different types of ridging are influenced by the strength in the horizontal gradient of the geopotential ($\nabla_p \phi$). The thin black

contours in Fig. 3 show that these gradients are much stronger within the 30°S and 50°S latitudinal belt, than north and south of it, during the ridging process, so that the geostrophic flow (and by approximation the flow) is strongest there, thus informing the location of $K_e$ (Fig. 3).

To identify the $K_e$ centre more easily, we label these in red in Figs 3b and f as centre I, II and III. Centres I and II occur west and east of the positive geopotential height anomaly that arrives in South African domain as a component of the wave

packet that passes over the country as ridging occurs (see Ndarana et al., 2022). Energy centre labelled III occurs east of the negative geopotential anomaly that in turn is located to the east of the positive geopotential anomaly. The magnitudes of the energy centres characterise the first difference between the two types of ridging. All three centres, in particular I and II are



stronger for Type-S events compared to Type-N ridging. This may be a consequence of $\nabla_p \phi$ induced by Type-S events being stronger, thus leading to a more intense perturbation flow field, as discussed above. The other key observation to be made in

Fig. 3 is that, for both types of ridging events, energy centre I maximises at the inception of the ridging process, whilst energy centre II attains its maximum values, as the ridging process matures, as indicated by the SAOH having extended across the South African domain. The latter is largely due to the fact that the ridging process induces progressively stronger $\nabla_p \phi$ so that the perturbation flow increases and, hence, $K_e$ . This also explains the seasonal variations in $K_e$ as shown in Fig. 4, in which the yellow to brown shading shows values of $K_e$ up to 65 m$^2$ s$^{-2}$, as in Fig. 3 and the grey shading showing values in excess of

this value. It is clear from this figure that there are strong seasonal variations in $K_e$ with magnitudes that exhibit the largest values during the winter months. This comes from the fact that the ridging induces stronger low level geopotential anomalies from DJF to JJA (Ndarana et al., 2022).

This discussion suggests that there is no clear sign of downstream development at the lower levels of the troposphere during the ridging process in the South African domain. This follows from the fact that at least for Type-S cases, energy centre III

weakens as centre II becomes slightly stronger. This will be clarified further in the following sub-section by considering the ageostrophic geopotential fluxes, which, as discussed in Section 2, define the downstream development process.

### 4.3 Low-level $K_e$ transfer processes

The discussion in Section 4.1 suggests that downstream development of $K_e$ is predominantly an upper level process, but it falls short in demonstrating whether the process exists at the surface or not, because it merely shows the variations in the magnitudes of $-\nabla \cdot (\mathbf{v}_a \phi)$. Furthermore, the evolution of the $K_e$ discussed in the previous section is also not conclusive on this question. Therefore, to address this issue further, the evolution of $\partial_t K_e$ , relative to $K_e$ , together with the convergence/divergence of the flux terms in Eq. (1) and (2) at the surface are discussed in this section. This is done with the

aide of Fig. 5, which shows the composite evolution of the patterns of these terms at t = 0 hours. The top panels (Figs 5a and b) show that, broadly, $K_e$ increases over eastern half of the centre and decreases in rear end of it. This is to be expected because as the ridging process evolves, the surface $K_e$ propagates eastward, as discussed in the previous section. To assess whether this behaviour is characteristic of downstream development, we first consider the ageostrophic flow that is induced by the ridging process. If we define the axis of a ridging high as the curve where $\partial_y P = 0$, where $P$ is the MSLP, then this curve

would follow or trace all the points within the high pressure system where the pressure is a maximum. Equatorward of the axis defined in this manner, particularly as the ridging system curves around South Africa, the ageostrophic flow always points inland; with implications for moisture transport (Ndarana et al., 2021a). Poleward of this axis, the ageostrophic flow has a south-western orientation, right across the South Atlantic Ocean, regardless of whether it flows across a surface trough or ridge axis. Changes from t = 0 hours (Figs 5c and d) to t = +24 hours (not shown) and beyond, show that this behaviour of the

ageostrophic flow (together with its divergence/convergence patterns – the shaded regions in Figs 5c and d) is maintained right





through the evolution of the ridging process. This orientation of the ageostrophic flow relative to the perturbation geopotential height fields is the second clue that downstream development might not exist at the surface, as it appears to fail to comply with the patterns reviewed in Section 2. This applies to both types of ridging.

The above has profound implications for energy transport. Combining the behaviour and structure of the ageostrophic flow and the patterns of $\phi$, as well as those of $\mathbf{v}_a\phi$ and its convergence/divergence that are very different from those expected for downstream development emerge. For Type-N energy, growth that is associated with the first energy centre appears to first be influenced by cyclonic ageostrophic fluxes in the mid-latitudes that converge (i.e. where $-\nabla \cdot (\mathbf{v}_a\phi) > 0$) near the Greenwich Meridian (Fig. 5e). These fluxes diverge from just south of 60°S, and between 30°W and the Greenwich Meridian. This pattern is very different from the one induced by Type-S ridging highs. In that case, the fluxes are oriented in a south-westerly direction (Figs 5f), just south of energy II. This pattern suggests that energy is transported from energy centre II towards I, which is opposite to what is expected during downstream development. Energy centre III, located just west of the Indian Ocean high, also grows from ageostrophic geopotential fluxes that have a north-easterly configuration. They appear to originate from a flux divergence region with the positive MSLP anomaly that is induced by the ridging process, and not by the transfer of energy from energy centre II. All of the above is not indicative of energy transport across a trough (ridge) where the flow is subgeostrophic (supergeostrophic), so that the energy centre located downstream grows at the expense of the centre immediately upstream of it. This confirms observation made in the previous section that there is no evidence of downstream development at the surface during the evolution of ridging highs.

Since $K_e$ is always positive, $\mathbf{V}K_e$ will clearly be oriented as the surface circulation, which is shown by the arrows in Figs 3 and 4. In a baroclinic wave, the effect of this energy flux is to distribute $K_e$ from the rear end of the energy centre to the front of it. In this case however, the surface circulation transport energy in south-easterly (southerly) direction during the evolution of Type-N (Type-S) events, just west of the surface MSLP anomaly (i.e. in the rear end of the leading edge of the ridging high), within energy centre I. In the Southern Hemisphere, it is well known that the flow is directed towards the southeast ahead of the cold front but towards the northeast behind it, ahead of the ridging high. This means that the effect of the cold front is to grow energy centres II and III.

## 4.4 Evidence of downstream development at the upper levels

As a first step to presenting evidence of downstream development aloft during the evolution of baroclinic waves that are associated with ridging high pressure systems, we show graphs of composite maximum $K_e$ as functions of time-lag in hours in Fig. 6. Therefore the green, blue and red curves in Fig. 6 respectively represent the evolution of the maximum of the composite eddy kinetic for centres I, II and III, labelled from west to east as schematically represented and highlighted by the thick oval shapes in Stage 2 in Fig. 1. The dots on each curve represent the times at which the centre of each of the components of the baroclinic waves enter and exit the South African domain (bounded by 10°E and 40°E) as they propagate east. These graphs are similar to Orlanski and Sheldon (1995) (cf. the top panel of their Fig. 1) and are able to succinctly capture the





evolution of $K_e$ and variations of this evolution with season and with type of ridging as the various components pass through

the domain.

     We first recall that t = 0 hour on the x-axes of Fig. 6 is the composite time at which the riding process is initiated and this

is used as reference point for the discussion. The top panels of Fig. 6 show that in the case of Type-N ridging events the growth

of energy centre II saturates during the 24 hours leading up to the initiation of the ridging process, whereas energy centre III

reaches its largest values during the first 24 hours after the ridging process has begun. Centre III enters the South African

domain first as it is located downstream. The dots on the red curves show that $K_e$ associated with this centre reaches a

maximum and dissipates when that component of the wave is over the Indian Ocean; meaning that it grows whilst it is in the

South African domain. This is indicated by the  red dots being located on the increasing side of the red curve. The dots on the

blue curves similarly show that centre II dissipates in the South African domain. It grows whilst it is over the South Atlantic

Ocean and peaks just before it crosses the western boundary of the domain, marked by the 10 °E longitude line.  Energy centre

I enters the South African domain substantially later, and only briefly beyond t = +48 hours, when the ridging process has

completed in most cases.  These features are common to all seasons. The variations between seasons are observed for the

relative maximum $K_e$ values that centres II and III attain. During DJF the $K_e$ associated with centre II is lower than that

associated with centre III downstream, they are of comparable magnitude during MAM  and during the winter months centre

III attains the largest $K_e$ value.. Changes that lead to the summer structures start occurring in spring.

     The evolution of the $K_e$ associated with the most important components of the Type-N baroclinic wave is reminiscent of

Life Cycle 1 (LC1; Thorncroft et al., 1993; Hartmann and Zuercher, 1998). In contrast, the $K_e$ of the waves associated with

Type-S events (shown on the bottom panels of Fig. 6) taper off in a similar fashion to Life Cycle 2 (LC2; Thorncroft et al.,

1993; Hartmann and Zuercher, 1998). These latter baroclinic waves have longer life spans (Ndarana et al., 2022), which is

consistent with the behaviour of the $K_e$ discussed here. The structure of Type-S energy centre II, in particular, starts off in an

LC2-like manner (cf. Fig.4 in Thorncroft et al. 1993) during the summer to become more like LC1 during the winter months.

This is consistent with the fact that an upstream and southwest jet streak develops during the winter (Ndarana et al., 2022),

thus introducing anticyclonic barotropic shear, so that the waves that might have been breaking cyclonically, as a result of

being located on the poleward side of the upstream jet, now break anticyclonically. To support this argument about the change

in $K_e$ structure, Hartmann and Zuercher (1998) showed by introducing cyclonic shear, LC1 becomes LC2, and even though to

the best of our knowledge, no study has shown the opposite, it is plausible that an addition of anticyclonic barotropic shear

may change LC2 to LC1. Note that Type-S energy II starts growing in the South Atlantic Ocean but reaches its maximum

values in the South African domain. The structure of Type-S energy III is also very different from that of Type-N. During DJF

and MAM it tapers off, in LC2 like fashion as energy centre II but becomes flat and lacks growth in winter. Note however that

the long lived nature energy centre III is the result of a completely different process from that influencing centre II. This will

be further explored below.  In spring energy centre III grows in the South African domain, and saturates over the South West

Indian Ocean, where it subsequently dissipates.





### 4.5 Baroclinic conversion and upper-level downstream development


We now consider the processes that underpin the evolution of $K_e$ aloft during the evolution of the two types of ridging highs that was discussed in the previous subsection. A convenient point of departure for the discussion of this section is the growth and decay of energy centre II (blue curves in Fig. 6). As this energy centre is located at the inflection point that is situated

between the upstream ridge and trough immediately downstream, there is strong upper level convergence (Ndarana et al. 2022), and hence downward motion (i.e. $\omega > 0$) associated with it. Recall also that $\phi$ decreases as a function of height from the ridge to the trough axis, implying that $\alpha = -\partial_p \phi < 0$, so that $-\omega\alpha > 0$. This, according to Eqs (1) and (2), represents the conversion from eddy available potential energy to $K_e$. Most of the growth observed for energy centre II is caused by this conversion as shown by the Hovmöller diagram in Fig. 7. For Type-N events during all seasons, as well as during the summer

and spring in the case of Type-S, baroclinic growth is confined within the region, where the meridional component of the perturbation flow, $v$, is positive within energy centre II. The top panels of Fig. 7 show that the baroclinic conversion increases whilst the energy component is propagating over the South Atlantic Ocean and maximises before energy II enters the South African domain. Therefore this explains the $\partial_t K_e > 0$ of the blue curve in Fig. 6. This baroclinic conversion also increases from DJF to JJA, which explains the progressively higher values of $K_e$ as clearly shown in Fig. 6. It also explains the higher

$K_e$ values associated with Type-S during JJA and SON. Note also that during the winter months substantial amounts of baroclinic conversion occur for energy centre I during Type-S evolution (Fig. 7g), hence the higher values during this season compared to Type-N (compare the green curves in Figs 6c and g). Regardless of season and type of ridging associated with the baroclinic waves, the growth from baroclinic conversion occurs upstream as discussed in the recent review of Rossby wave packets (Wirth et al., 2018). However, this conversion coincides with downward motion as it is located east of an upper ridge

axis so that latent heat release would not be expected, and thus no divergent amplification (Wirth et al. 2018).

$K_e$ in a baroclinic wave grows in one of two ways and by means of two processes, namely baroclinic conversion from eddy available potential energy and downstream development (Orlanski and Katzfey, 1991). Fig. 7 shows that there is no baroclinic contribution to the growth of energy centre III. Therefore it can only grow at the expense of energy centre II, by means of latter process which is presented in the first instance as Hovmöller diagrams of $-\nabla \cdot (\mathbf{v}_a \phi)$ shown in Fig. 8 (see Eq. 2). As illustrated

by this graph, there are large differences between Type-N and Type-S events and large seasonal variations, in keeping with other SAOH metrics (Sun et al. 2017). By comparing the top panels of Figs 7 and 8, and also taking Fig. 6 into consideration, it is clear that for the Type-N events, the baroclinic conversion takes place before energy II component of the baroclinic wave enters the South African domain. The conversion takes place at the rear end of the energy centre, whilst the energy sink, $-\nabla \cdot (\mathbf{v}_a \phi) < 0$, occurs downstream and dominant when energy centre II propagates past the South African domain. This means

that the decrease in energy centre II ($K_e$ of the blue curve in Fig. 6) is caused by this divergence of the ageostrophic geopotential flux. The top panels of Fig. 8 also show that at the rear end of energy III there is an energy course characterised by $-\nabla \cdot$





$(\mathbf{v}_a \phi) > 0$, which begins to occur at this component of the wave enters the South African domain, leading to $K_e$, according to Eqs (1) and (2) as shown in Fig. 6. The ageostrophic geopotential flux convergence is a maximum during MAM (Fig. 8b), so that $K_e$ saturates at higher values during this season compared to the others (Fig. 6b).

The bottom panels of Fig. 8 show that the situation is different for Type-S events. For one thing, during DJF (Fig. 8a), the strongest ageostrophic geopotential flux energy sinks and sources are located downstream from the South African domain. The weaker but elongated $-\nabla \cdot (\mathbf{v}_a \phi) > 0$ structure is in agreement with the fact that the red curve in Fig. 6b is not as steep as its Type-N counterpart in Fig. 6a, and also explains why energy centre III is longer lived for Type-S events. Similar arguments apply to MAM, whilst growth of energy III in Fig. 8f is not clearly defined as indicated also in Fig. 6. The structure

of these sources and sinks in Fig. 8h are very similar to those associated with Type-N and there is evidence of this similarity in the structure of the evolution of the $K_e$ in Fig. 6h.

     Finally, to explain the processes that underlie the structures of the energy sources and sinks shown in Fig. 8 we make use composite fields (Fig. 9), shown only for the time lag t = 0 hours because they are similar for the other time steps as the ridging process evolves. The top panels (Figs 9a and b) show $\partial_t K_e$ (shaded) and $K_e$ (hatched), middle panels (Figs 9c and d) show

the ageostrophic flow and its divergence/convergence and in the bottom panels (Figs 9e and f) the ageostrophic geopotential fluxes and their divergence/convergence are presented. All the fields are at 250 hPa. The patterns of $\partial_t K_e$ shown in the top panels are contributed to, amongst other processes, by $-\nabla \cdot (\mathbf{V} K_e)$ that acts to translate the energy centres eastward (Orlanski and Sheldon 1995) and $-\nabla \cdot (\mathbf{v}_a \phi)$ which, as previously discussed, describes the growth of energy by means of the downstream development process. We focus on the latter. The middle panels (Figs 9c and d) show that there is

supergeostrophic and subgeostrophic flow across the ridge (red curve marked R in Fig. 9c) and trough (black curve marked T), respectively. This means that across ridge (trough) axis the ageostrophic flow is directed downstream (upstream). A brief review of the flow involved in baroclinic waves is provided in Ndarana et al. (2022). The direction of the ageostrophic flow just described, combined with the sign of the $\phi$ across the ridge (where $\phi > 0$) and trough (where $\phi < 0$) axes lead to a radiation of energy from an upstream energy centre to one immediately downstream. For the two types of ridging highs

considered in this study, this flux is strongest across the trough axis (see bottom panels of Fig. 9), so that the most significant downstream development occurs from energy II to energy III. This is true for the baroclinic waves that are associated with both types of ridging, but more so for Type-N, as shown by the discussion of Fig. 8 above. This is quite different from what was found for COLs in the South African domain (Ndarana et al. 2021b) and for strong COLs in the Southern Hemisphere (Pinheiro et al., 2021) A more detailed comparison of these two downstream development regions is presented in the next

section.

## 4.6 Comparison with COL downstream development

We now update Fig. 1, in a geographical context, based on the above discussion and what is currently known about the COL

downstream development processes in the South Atlantic Ocean/South African sector. The differences between what has been





observed in this study and the Ndarana et al. (2021b) findings for COLs warrant a comparison between of the two types of regimes of downstream development found in South Atlantic Ocean/South African sector. The left panels of Fig 10 schematically summarise the Ndarana et al. (2021b) observations (drawn from their Figs 7 and 8) and the right panels show the findings of the current study, for both Type-N and Type-S ridging events. For both downstream development regimes, $K_e$

II is located between two jets streaks, one that is located upstream relative to it, which we call the upstream jet (UJ), with the other situated downstream; the downstream jet (DJ). Only Type-N events have a clearly defined UJ. Note that, whilst there are large differences between the fields associated with Type-N and Types-S ridging events, in as far as intensity and seasonality are concerned, the general characteristics associated with downstream development associated with them are similar but differ significantly from that associated with COLs. We list these differences as bullet points for clarity:


1.  The COL UJ propagates in a south-easterly direction by the distribution of zonal momentum from the entrance to the exit of the jet streaks (Ndarana et al., 2020) for the COLs case (see left panels of Fig. 10). This causes the deformation of the geopotential heights so that the trough approaching South Africa has a pronounced northwest/southeast

410         orientation tilt. This deformation is more clearly seen when viewed from a potential vorticity perspective (Ndarana et al. 2021b), in terms of which anticyclonic Rossby wave breaking (RWB) is defined (Thorncroft et al., 1993). What contributes to this is that the COL trough appears quasi-stationary, relative to the mid-latitude jet streak, so that $K_e$ I eventually catches up with $K_e$ II. The UJ observed during ridging highs does not propagate, in contrast to its COL counterpart. What this leads to is that the ageostrophic geopotential fluxes across the ridge axis (black line in Fig. 10)

415         associated with COLs are stronger than those in the upper level baroclinic wave that is associated with ridging high pressure systems.

2.  The DJ associated with ridging highs is much stronger and much larger in spatial extent than that associated with COLs. It's thermally direct transverse circulation at its entrance therefore makes a more significant contribution to the

420         downward motion west of the trough axis, thus leading to higher baroclinic conversion.

3.  Points 1 and 2 have profound implications for the differences in the evolution of $K_e$ that is associated with COLs from that associated with baroclinic waves that have been observed aloft during ridging high pressure systems. The COL eddy energy centre I grows from baroclinic conversion from eddy available potential energy. In the case of ridging

425         high baroclinic waves, the bulk of the baroclinic conversion is located downstream of the ridge axis. Because the supergeostrophic flow is weaker than it is for COLs, the ridging high energy centre II grows from this conversion, rather than from downstream development from energy centre I, as it is the case for COLs. This is a direct effect of the fact that the UJ is quasi-stationary for ridging high upper air waves, relative to the one observed for COLs.





4.    Downstream development during the evolution of COLs occurs across the ridge axis, as the waves break
anticyclonically (Ndarana and Waugh, 2010; Reyers and Shao, 2019; Barnes et al., 2021) and occurs across the trough
axis during the evolution of baroclinic waves that are associated with ridging highs. Thus, COL energy II grows at
the expense of energy I, whereas, ridging high energy III grows at the expense of energy II.

## 5. Concluding remarks

Using ERA-5 reanalysis from 1979 to 2020, this study employs the local energetics framework of Orlanski and Katzfey (1991)
to characterise the differences observed for Type-S and Type-N ridging high events in the South African domain (Ndarana et
al. 2022). Because of the significant differences between the flow at the surface and upper levels during the evolution of the
ridging high process, the energetics diagnostics considered in this study are not integrated in the vertical. This was a necessary
consideration in order to characterise the differences between lower and upper level dynamical processes during ridging highs.
By so doing it was shown that there is no evidence of downstream development at the surface during the ridging process. Also
the differences in the intensity of the eddy kinetic energy density between Type-N and Type-S ridging is influenced by the
strength of the perturbation flow that each ridging regime produces. The flow associated with the latter is stronger and so is
the associated eddy kinetic energy. There are significant seasonal variations, in keeping with other metrics associated with
SAOH pressure systems (Sun et al., 2017). The eddy kinetic energy at the surface increases significantly from the summer to
the winter months.

In contrast, at the upper levels there is clear evidence of downstream development. The evolution of the associated
baroclinic waves begins with three eddy kinetic energy centres. For referencing purposes, these are called centres I, II and II
and are located west of the ridge axis, between the ridge and trough axes, and downstream the trough axis. At the point of
initiation of the ridging process, the ridge is located near the Greenwich Meridian. Centres I and II, in particular the latter;
develop and grow from baroclinic conversion from eddy available potential energy to eddy kinetic energy for the most part.
The fact that these are located at inflection points means that there would be some influence of the ageostrophic geopotential
fluxes, but the intensity of energy I relative to energy II clearly indicates that the eddy kinetic energy density in the latter cannot
come from the former. It may then be concluded that it is generated by baroclinic conversion. There is little evidence or none
at all of baroclinic conversion that is associated with energy centre III. This centre almost entirely develops and grows from a
radiation of energy from centre II across the trough axis. Chang (2000) found examples of these downstream development
processes located in the manner described in this study in the South African domain.

There are clear differences in the intensity of the energy densities between Type-N and Type-S and their generation
processes. These differences are observed during all seasons. The baroclinic conversion associated with Type-S events is much
stronger, leading to a more intense centre II. However, during the winter months the downstream development during the
waves associated with Type-S events is weaker and less clearly defined, than in the Type-N case.



The baroclinic wave dynamics findings in the study may then be summarised in this manner: Baroclinic waves that are associated with ridging high pressure systems develop from baroclinic instability in the mid-latitudes over the South Atlantic
Ocean, as they propagate east. The development of the trough that eventually induces vertical motion over South Africa occurs via downstream development. This downstream baroclinic wave development is different from that which is associated with COLs both in terms of orientation and also terms of which aspect of the wave develops from baroclinic conversion. Precisely why these differences exist remains an open question and a subject of further study. It is also apparent that characteristics of downstream development in the South African domain and surrounding oceans is greatly influenced by the dynamics of the
jet streaks. In particular the movement of the upstream jet streak. As such, another open question is why does this jet propagate so much faster during the evolution of COLs compared to that of upper air baroclinic waves that are associated with ridging highs in general? Also, contrasts between the jet streak configurations associated with the two types of ridging events suggest that barotropic shear influences the evolution of the energy centre that develops from downstream development. This hypothesis is based on classic idealised studies of baroclinic life studies (e.g. Thorncroft et al., 1993; Hartmann and Zuercher,
475 1998).

*Author contributions:* TN, LL and TS conceptualised the research, conducted the analysis and LL prepared the graphs, all authors contributed to the writing and editing the manuscript

*Competing interests:* There are no competing interests

*Acknowledgements:* This work is part of a broader project that, whose aim is to understand that the underlying processes of the important synoptic systems in the South Atlantic Ocean/South Africa basin. The authors would like to the Water Research Commission (Grant number: C2020/2021-00653) for its support.

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




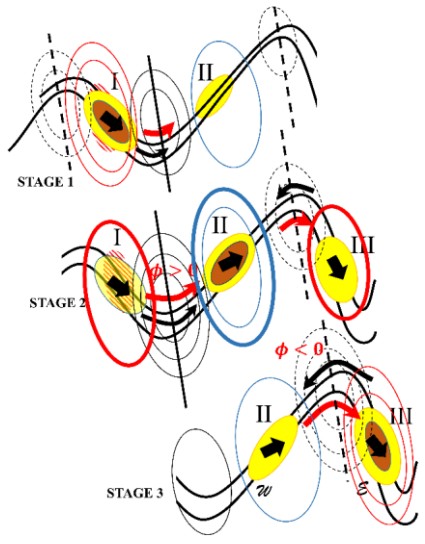


**Figure 1:** Schematic representation of a propagating baroclinic wave (black curves) together with its evolving eddy kinetic energy ($K_e$) density centres (yellow and brown shaded oval shapes and labelled in Roman numerals I, II, and III - the brown shade represents increasing strength of eddy kinetic energy) and geopotential perturbation ($\phi$) patterns (oval shapes with no shading). The thin solid (dashed) contours represent areas in the wave where $\phi > 0$ ($\phi < 0$). The black curved arrows represent the ageostrophic flow ($\mathbf{v}_a$) and the red curved arrows represent the ageostrophic geopotential fluxes, ($\mathbf{v}_a\phi$). The solid (dashed) straight line represent the ridge (trough) axis. The increasing size of the arrows and increasing number of contours representing ($\mathbf{v}_a$) and ($\mathbf{v}_a\phi$) in the oval shapes that represent the $K_e$ and $\phi$ represent increasing intensity of these fields. The red (blue) unshaded oval shapes represent areas in the baroclinic wave where the perturbation meridional velocity field is negative (positive). [Adapted from Orlanski and Sheldon (1995). It also draws from the findings of Ndarana et al. (2021) and Pinheiro et al. (2021) for COLs in the South Atlantic Ocean/southern African sector and Southern Hemisphere, respectively.]






**Figure 2:** Vertical profiles of various diagnostics in Eqs (1)associated with Type-N (Type-S) ridging events averaged in the domain bounded by 30°S and 60°S to the north and south, and 30°W and 60°E to the east and west, from $t = 0$ hours to $t = 48$ hours for events with a 48 hour duration. The green and blue curves represent negative values for Type-N and Type-S ridging and the red and black curves represent positive values for Type-N and Type-S ridging, respectively. The fields shown are the (a) eddy kinetic energy, (b) zonal perturbation velocity component, (c) meridional perturbation velocity component, (d) eddy kinetic energy tendency, (e) ageostrophic flow convergence, (f) the vertical velocity, (g) specific volume, (h) baroclinic conversion and (i) ageostrophic geopotential flux convergence.



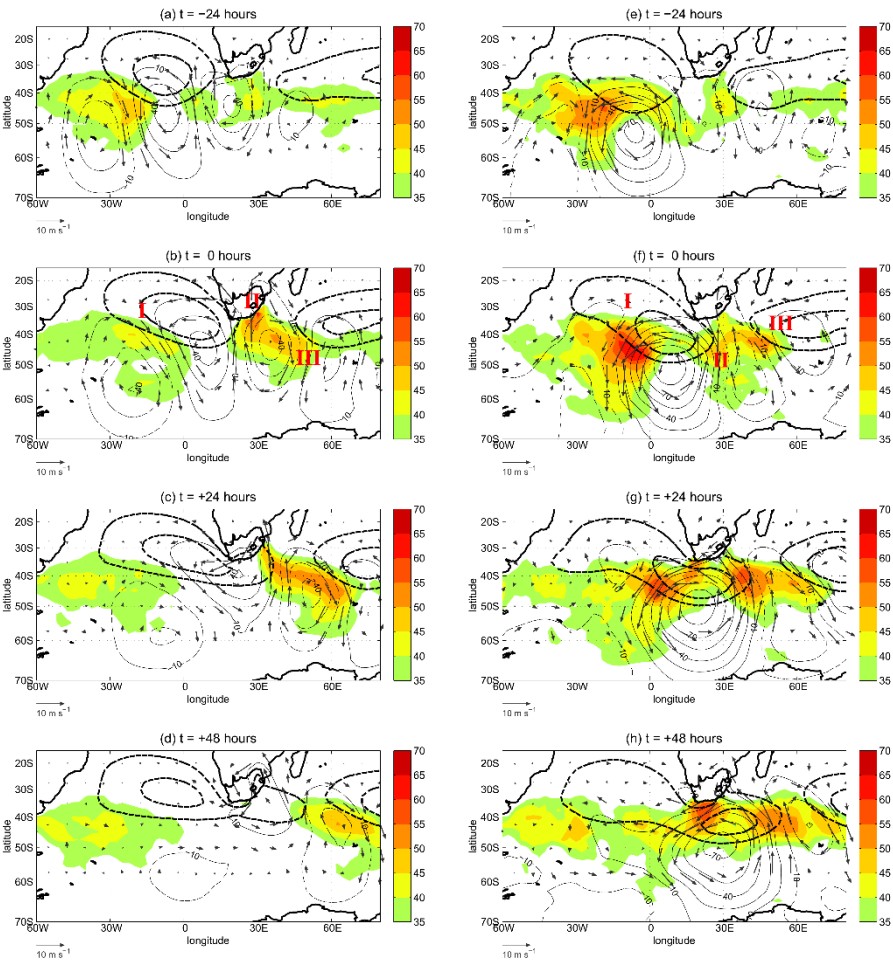

**Figure 3:** Time-lagged composite mean fields of mean sea level pressure (thick dashed contour), 1000 hPa eddy kinetic energy
(shaded), positive (negative) 1000 hPa geopotential height perturbations represented by thin black solid (dashed) contours
plotted at 15 m² s⁻². The grey arrows represent the perturbation geostrophic flow $v \approx f^{-1}k \times \nabla_p \phi$ at 1000 hPa for Type-N
(left panels) and Type-S (right panels) events. The eddy kinetic energy centres are labelled in red Roman numerals from west
as I, II and III shown in (b) and (h). The thick dashed contours are the 1016 and 1020 hPa MSLP contours. The composites
are plotted from (a, g) t = -24 hours to (f, l) t = +48 hours, in 24 hour time intervals.



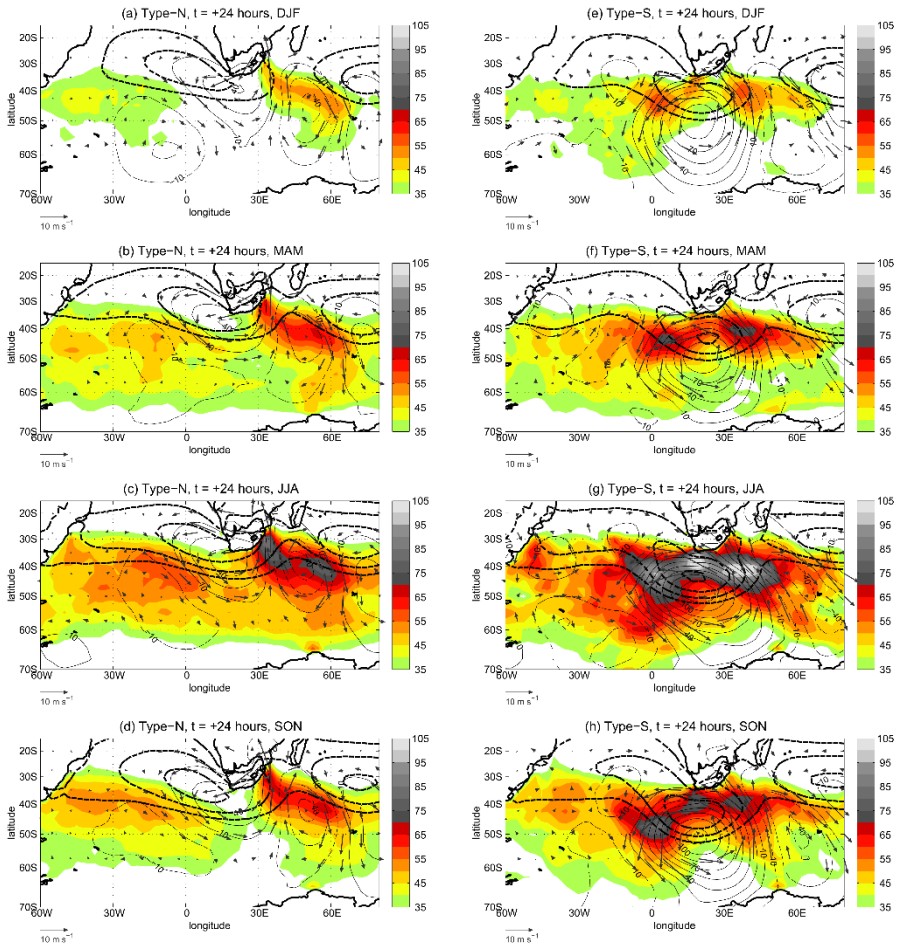


**Figure 4:** Composites of the variables shown in Fig. 3 but only for t = + 24 hours and for (a,e) DJF, (b,f) MAM, (c,g) JJA and (d,h) SON. The left and right panels are for Type-N and Type-S events, respectively. The colour bars are presented such that the yellow-brown represent values of $K_e$ up to 65 m$^2$ s$^{-2}$ as in Fig. 3, so that the grey shading represent values of $K_e$ that are higher than this value.



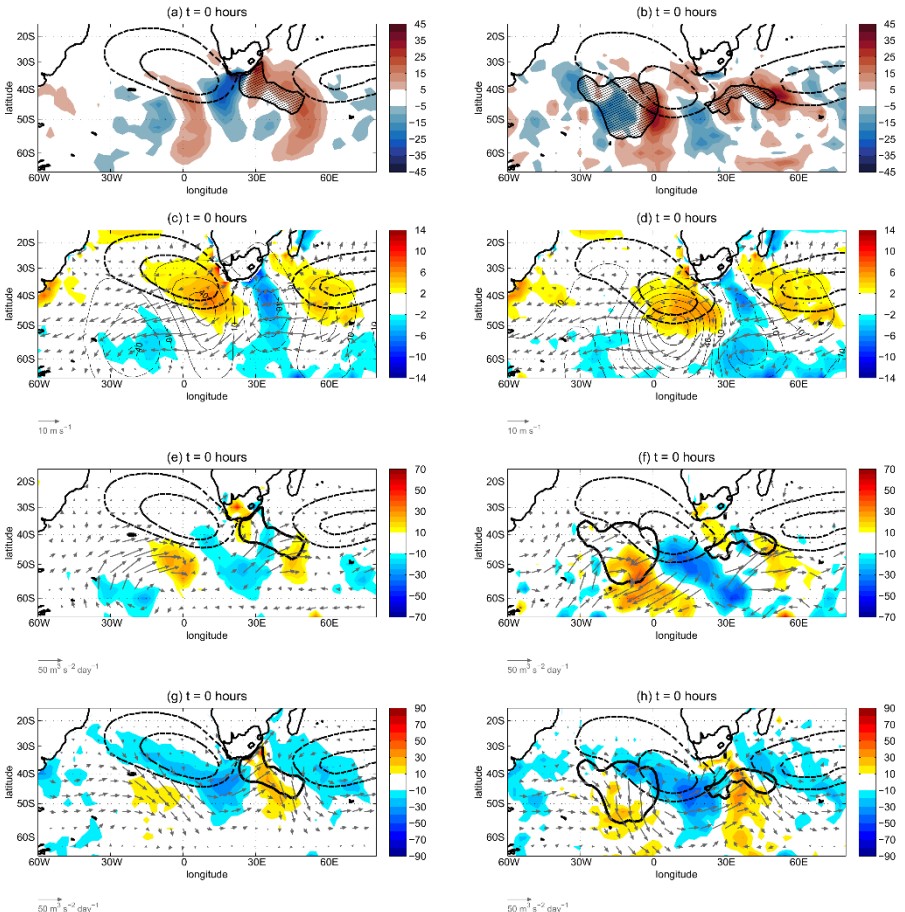


**Figure 5**: (a, b) Composite mean fields of $K_e$ (hatched region with thick black contour) and $\partial_t K_e$ (shaded). (c,d) Composite mean fields of the ageostrophic flow ($\mathbf{v}_a$) represented by grey arrows, its divergence at the surface represented by the shading and geopotential height anomalies represented by the thin black contours plotted in 15 m interval levels. (e,f) Composite mean ageostrophic geopotential fluxes ($\mathbf{v}_a \phi$) represented by the grey arrows, their divergence ($-\nabla_p \cdot (\mathbf{v}\phi)_a$) represented by the

shading. The thick red contour represents $K_e$. (g, h)Composite mean fields of eddy kinetic energy fluxes ($\mathbf{V}K_e$) represented by the grey arrows and their divergence ($-\nabla_p \cdot (\mathbf{V}K_e)$), represented by the shading, with the $K_e$ reproduced as in (c) and (d). All the fields shown are at 1000 hPa and the thick dashed contours are the 1016 and 1020 hPa MSLP contours. The thick dashed black contours show represent the 45 m$^2$ s$^{-2}$ contours of $K_e$ for Type-N (Type-S) ridging events. The left and right panels are for Type-N and Type-S events, respectively. The composites shown are for t = 0 hours and DJF at 1000 hPa.






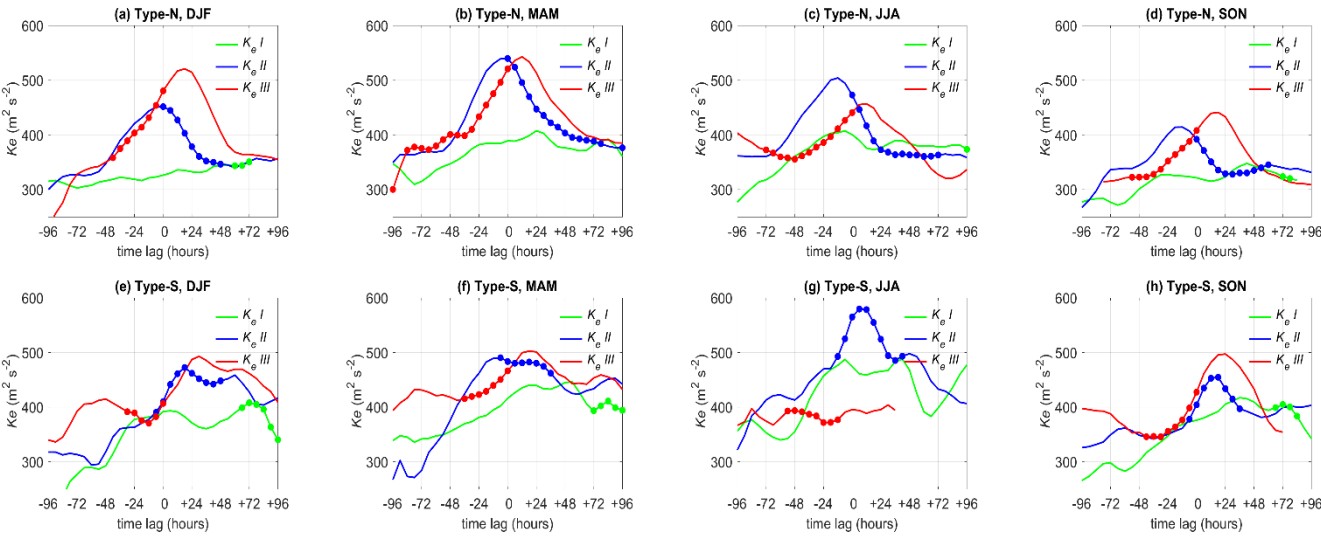

**Figure 6**: Curves of composite maximum eddy kinetic of centres I, II and III (see Stage 2 in Fig.1) that passed through the South African domain bounded by 10°E and 40°E as a function of time-lag in hours. The dots on the curves represent the periods during which the components of the baroclinic wave passes over the South African domain. The green, blue and red curves represents the maximum kinetic energy for centre I, II and III. The top (bottom) panels are for Type-N (Type-S) events from DJF to SON.







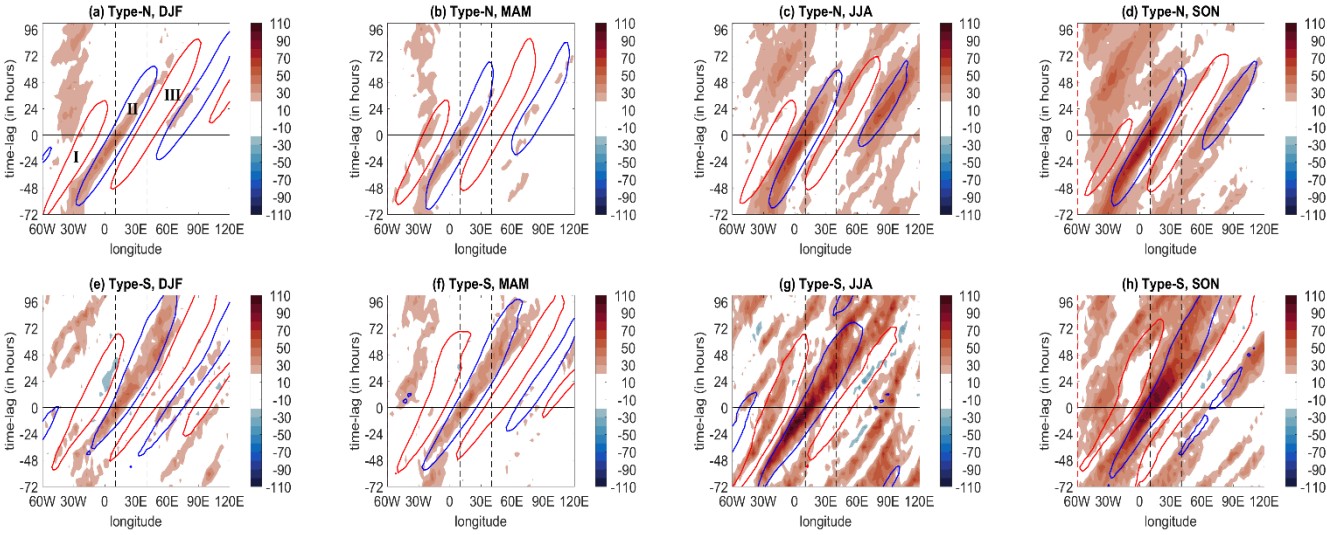

**Figure 7:** Hovmöller diagrams for $\omega\alpha$ (shaded) and $v$ for Type-N (top panels) and Type-S (bottom panels) for (a, b) DJF, (c, d) MAM, (e, f) JJA and (g ,h) SON. The red (blue) contours represent negative (positive) values of $v$, consistently with Fig. 1 and Fig. 5 and $K_e$ centres I, II and III are labelled in Fig. (a). Only the $\pm 4$ m s$^{-1}$ contours are shown. The red dashed line represents the location of the 60°\$W longitude, and the black dashed lines represent the location of the South African domain that is bounded by 10°E and 40°E.









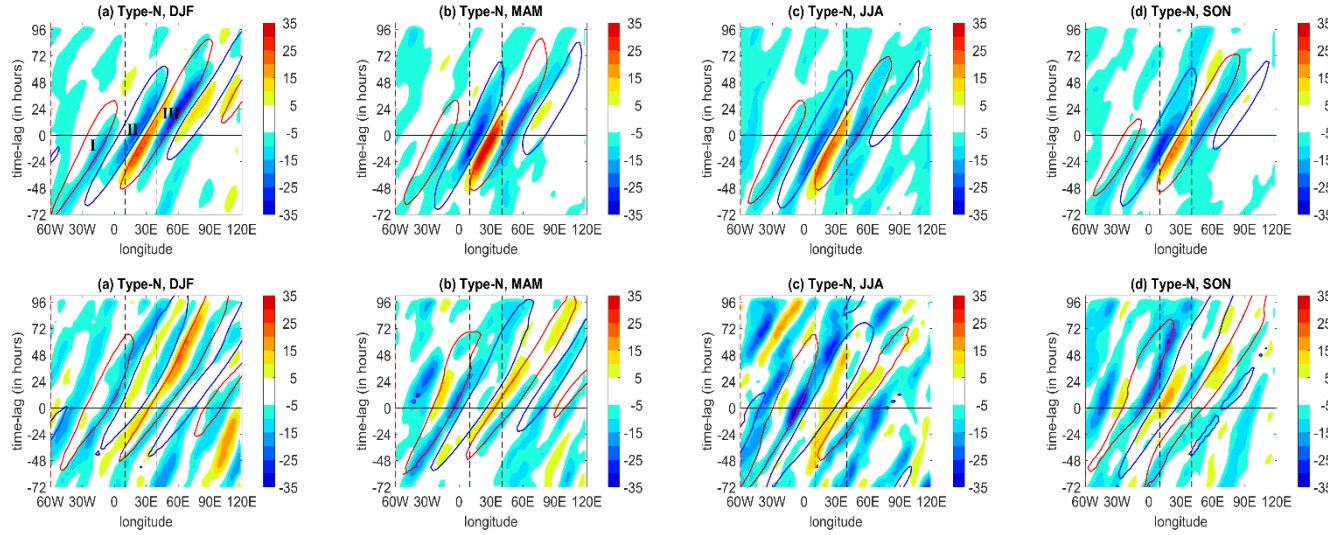


**Figure 8**: Same as Fig. 7 but for $-\nabla_p \cdot (v\phi)_a$.





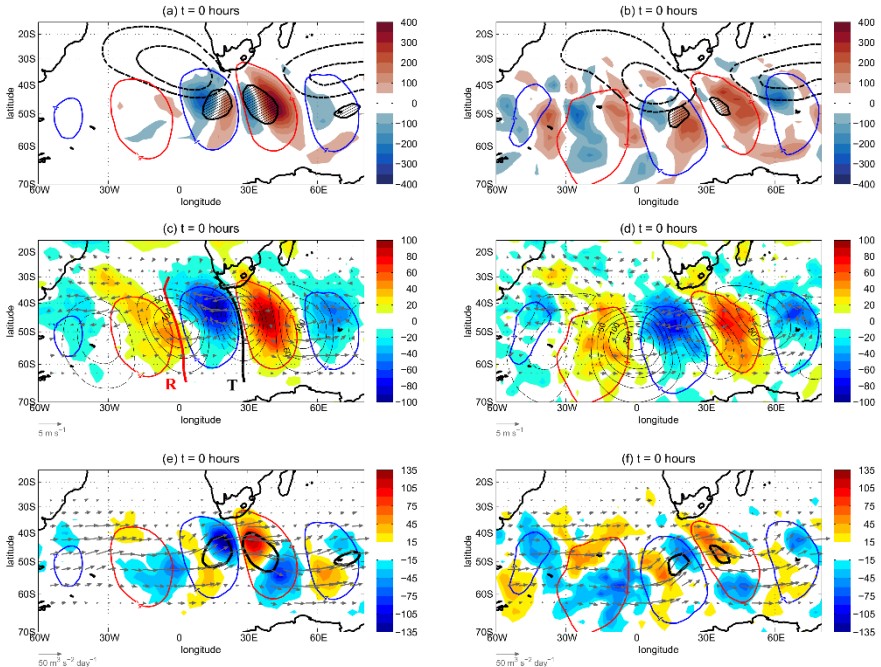

**Figure 9**: (a, b) Composite mean fields of $K_e$ (thick black contours) and $\partial_t K_e$ (shaded). (c, d) Composite mean fields of the ageostrophic flow ($\mathbf{v}_a$) represented by grey arrows, its divergence at the surface represented by the shading and geopotential height anomalies represented by the thin black contours plotted in 25 m interval levels. (e, f) Composite mean ageostrophic geopotential fluxes ($\mathbf{v}_a\,\phi$) represented by the grey arrows, their divergence $\left(-\nabla_p \cdot (\mathbf{v}\phi)_a\right)$ represented by the shading and $\partial_t K_e$ represented by the hatched regions. The hatched region represent areas where $-\omega\alpha > 0$. All the fields shown are at 250 hPa and the thick dashed contours are the 1016 and 1020 hPa MSLP contours. The thick dashed black contours show represent the 380 (390) m$^2$ s$^{-2}$ contours of $K_e$ for Type-N (Type-S) ridging events, and these are filled with hatches in the top panels. The left and right panels are for Type-N and Type-S events, respectively. The composites shown are for t = 0 hours.





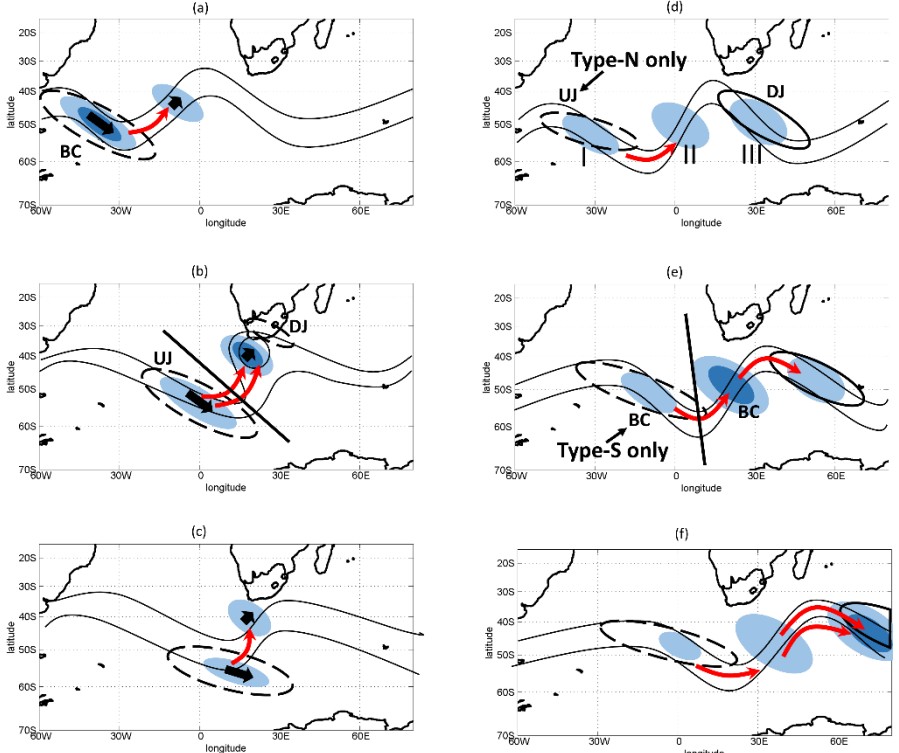


**Figure 10**: Schematics of the evolution of eddy kinetic energy centres (blue shaded oval shapes) associated with COLs (left panels) adapted from Ndarana et al (2021b) and ridging events. The darker blue indicate the intensity of the kinetic energy. The red curved arrows represent the ageostrophic geopotential fluxes and the number of arrows represent the strength of these fluxes. The middle panels represent the point at which the COL forms and the commencement of the ridging process. The solid

line represent the ridge axis and the black solid curves represent the geopotential height contours at upper levels. The thick dashed black oval shapes represent the jet streaks. Where baroclinic conversion dominates is marked by the letters BC. The right panels combines the downstream development processes and jet structure of both types of ridging high types and does not differentiate between the two, for purposes of comparing with COL downstream development.