# Peer review of "Downstream development associated with two types of ridging South Atlantic Ocean anticyclones over South Africa"

_Weather and Climate Dynamics, 2022_

## Referee Comment (RC3)

**Paper synopsis and general comments:**
Ndarana et al explores the dynamics and development of two types of ridging in the South African region. They look at all levels and approach the problem in an energy framework. Downstream development of the ridge occurs via the conversion of potential energy to eddy kinetic energy upstream of the ridge, and ageostrophic geopotential fluxes across a trough over South Africa more. Development does not occur at the surface, and dynamical differences in between ridging and cutoff lows are discussed.

Overall, there is a lot of very interesting analysis and though-provoking findings. However, there are many instances throughout the paper where the text could be tightened and trimmed as long paragraphs, long sentences, and occasionally unrelated information make it difficult to follow the arguments and overall aim of this paper. Nevertheless, once these comments are addressed, this paper could be a useful contribution to Weather and Climate Dynamics.

**Major Comments:**
1) Much of the text, and particularly the introduction could be trimmed to include only the necessary information. Currently, is difficult to resolve the motivation for the study. The lack of clear motivation makes it hard see the context for the background mathematical theory in section 2 and the methods in section 3. The methods section was hard to follow and could benefit from editing, and did not seem to follow logically from the equations in section 2. Without a clearer understanding of what you are doing and why, it became difficult to follow your results, or connect them to the theory in section 2. Careful editing throughout to only provide information relevant to addressing a clear aim of the study would greatly improve your manuscript.

2) Currently, it is difficult to disentangle your new results from those in Ndarana et al., 2021. There is clear and obvious overlap between your current work and how it relates to cutoff lows. While you make qualitative comparison between the wave trains associated with cutoff lows and those with ridging highs in section 4.6, and find interesting differences, a more careful and quantitative comparison could be beneficial. Careful consideration of the motivation for this study will help decide whether further quantitative comparison is necessary, or whether the focus is very specifically on the dynamics of ridging highs, with comparison to cutoffs only necessary in the discussion.

**Specific comments:**

Line 27: I don't think you consider stratospheric conditions through this paper, could you comment on tropospheric influences either here or later? Or relate kinetic energy to changes to stratospheric conditions?

Lines 31 – 36: While it is a good point that there are other moisture sources, I feel it distracts from the main goal paper. It may be sufficient to simply state that the South Atlantic and Indian are amongst the important water sources and remove these lines.

Line 37: in the previous paragraph (specifically lines 29-30) I initially thought ridging highs were important for moisture in southern Africa because of the onshore flow changing moisture fluxes. Is this not the case? Or are changes to onshore flow less important than the association with ridges and cut off lows? If so, why look at ridges and not cut off lows, as previous work has done? Some clarification on these points would be welcome.

Lines 37 -55: this is a rather long paragraph that seems to have two separate themes. Would you consider breaking this into two paragraphs: the first (lines 37-46) discussing the importance of ridges to rainfall, and the second (from "One approach…" on line 46 onward) discussing the energetics framework?

Line 62: I am unclear what you mean by a downstream development perspective. Do you mean that the dynamical characteristics of these waves downstream from initiation is not well understood? Or that the weather induced downstream from these waves is not well understood? Some clarification would be welcome here.

Line 63: Do you mean here that COLs are an example of this downstream development? Or do you refer to development downstream of the COL? (i.e. what is developing downstream?). See also line 70.

Lines 64 – 67: There is a lot of complex detail in this sentence without much explanation. It may be better moved to section 2 where you explain in detail the relationship between geopotential fluxes, kinetic energy, and COL development.

Line 67: Is there a relationship between rainfall amount and whether a surface ridge is co-located with an upper COL? I do not have a good feel for why exploring the dynamics to ridges is relevant to the extreme rainfall discussed above.

Line 85: Do you mean downstream development troughs and/or ridges over south Africa? At the surface or upper-level? It would be helpful to have a bit more detail or an opening sentence that explains the context for this section.

Equation 2 – line 104: The subscript 'a' has not been defined. Does this mean 'ageostrophic'?

Line 105: Please add $\alpha$ is the specific volume to match Fig 2.

Line 116: It would be helpful to include a brief description of why these inflection points are the most important centres for eddy kinetic energy.

Line 148: Is a word missing at the end of this sentence? Assumed…? The wording is somewhat confusing here.

Line 150: I am unclear what you mean by 'natural fashion'. Or how this signals the ridges climatological presence through an event. Could you clarify please?

Line 152: I think 'the' is needed here, before 'time step'

Line 154: Are the perturbation variables calculated at each time step for each variable? It may help to specify.

Lines 147 – 162: There may be some repetition between these paragraphs. Please consider whether these paragraphs could be streamlined.

Line 156: It is unclear why you are referring to shifting COLs and it may distract from the methods that you do use in this paper. Please consider removing, or clarifying why you are comparing COLs to ridges (e.g. 'this previous study shifted COL centres...').

Line 157: 'shift' not 'shifte'

Line 162: A full stop appears to be missing here.

Line 176: 'diagnostics under consideration' is rather vague. Consider specifying e.g 'diagnostics from equations 1 and 2'. I also note that the diagnostics in Fig. 2 are not all components of the equations. Please consider either plotting only the most relevant components of the equations or, in section 2, provide a short derivation explaining why these components are most relevant for the equations.

Line 177: What do you mean by a negative ridge event? Is this a trough?

Line 187: It may help reader comprehension to refer back to the equations and detailed explanation in Section 2 (see also comment for line 176)

Line 195: I am not convinced that it is clearly an upper-level process. The baroclinic conversion appears to predominantly occur in the low or mid troposphere. Why is the upper-level convergence of ageostrophic flux is dominant? Scale analysis of the terms in equations 1 and 2 may be beneficial here.

Line 196: In Fig 2, the convergence of ageostrophic flux is noted as $u$, not $v$ as it is in the text and equations.

Line 231: I may be misunderstanding the structure of the subtropical South Atlantic Ocean High, but to my eye, I do not see a ridge extending from the southern Atlantic over South Africa, but two different wave trains both with high-pressure nodes over South Africa and low-pressure nodes over the southern Atlantic. Could you comment or clarify around the SAOH please?

Line: 233: It is not clear what 'this' is referring to at the start of the sentence. Are you referring to a seasonal variation in the gradient of the geopotential height perturbation? Given the strength of the seasonal variation in Fig 4, it seems more could be said about the seasonal variability of the kinetic energy – unless this is covered more extensively in Ndarana et al., 2022?

Line 247 – Doesn't the vertical profile in Fig 2 given an extension at the surface? Could you clarify why plotting components of the convergence term in Fig 5 helps show the impacts at the surface?

Line 263: Could you give more detail around how this process does not match the idealised version in Section 2? What about the other components of the kinetic energy equation 1? Particularly the other components of equation 2. Could they explain surface development?

Line 312: I believe this is the first time that Life Cycles 1 and 2 have been mentioned. It would help the reader to include some explanation of what it is. An explanation is particularly important as this whole paragraph and the lead in to the next section revolve around a comparison with LC1 and 2.

Line 331: Did you not start exploring the evolution and components of Ke in section 3?

Line 351: To make it easier to follow this paper, I recommend you refer back to equation 2 when you talk about baroclinic conversion and downstream development. It was not immediately clear you were talking about the first two terms of this equation. What about the third term (vertical geopotential flux convergence)?

Lines 448: With no downstream development at the surface, what is the implication for rainfall with these events? How is that different with what is found with COLs?

**Comments on figures**
a) Figure 2: The plots do not match the diagnostics in the caption.

b) Fig. 2: For easier comparison between positive and negative events of each North and South Type, please consider pairing colours to each type (e.g. North (south) is red (blue) with positive being solid and negative being dashed)

c) Fig 3 It is difficult to see the labels with red contours underneath. Please consider using a different colour to highlight the k